# Spatial Correlation Network and Driving Effect of Carbon Emission Intensity in China's Construction Industry

Zhenshuang Wang [1], Yanxin Zhou [1], Ning Zhao [2], Tao Wang [3,*] and Zhongsheng Zhang [1]

1. School of Investment and Construction Management, Dongbei University of Finance and Economics, Dalian 116025, China; zswang@dufe.edu.cn (Z.W.); zyx18512463589@163.com (Y.Z.); guochengkongjian@163.com (Z.Z.)
2. School of Finance, Dongbei University of Finance and Economics, Dalian 116025, China; ningzhao5@163.com
3. School of Public and Policy Administration, Chongqing University, Chongqing 400044, China
* Correspondence: wangtaothu@163.com

**Abstract:** To explore the spatial network structure characteristics and driving effects of carbon emission intensity in China's construction industry, this paper measures the carbon emission data of China's construction industry in various provinces from 2006 to 2017 and then combines the modified gravity model and social network analysis method to deeply analyze the spatially associated network structure characteristics and driving effects of the carbon emission intensity in China's construction industry. The results show that the regional differences of the carbon emissions of the construction industry are significant, and the carbon emission intensity of the construction industry shows a fluctuating trend. The overall network of carbon emission intensity shows an obvious "core-edge" state, and the hierarchical network structure is gradually broken. Economically developed provinces generally play a leading role in the network and play an intermediary role to guide other provinces to develop together with them. Among the network blocks, most of the blocks play the role of "brokers". The block with the leading economic development has a strong influence on the other blocks. The increase in network density and the decrease in network hierarchy and network efficiency will reduce the construction carbon emission intensity.

**Keywords:** carbon emission intensity; gravity model; social network; driving effect; regional difference; construction industry; block model

## 1. Introduction

Global warming has brought serious challenges to human survival and development, causing serious harm to the environment and being a serious challenge for the future development of mankind [1]. In recent years, carbon emissions have been increased annually. In 2017, the total carbon emissions of China accounted for 27.2% of the world's emissions, and China has become the largest carbon emitter in the world [2]. China's carbon emission reduction pressure is huge, and it is the first developing country to formulate and implement a national plan to deal with climate change, which plays an important role in the world's carbon emission reduction issue [3]. A large number of carbon emissions will bring great pressure to the ecological environment, leading to the change of a country's climate conditions, and then affect the production capacity, which is not conducive to sustainable economic development [4].

The construction industry is a typical energy-intensive industry with significant characteristics such as high energy consumption, high emissions and a low technical level. The energy consumption of the construction industry accounts for about 36% of the total global terminal energy consumption, which generates about 40% of the global carbon emissions [5]. To address the global climate change issue, China has proposed to achieve peak carbon emissions around 2030 and carbon neutralization by 2060 to realize the Sustainable

Development Goals (SDGs) [6]. Carbon emission reduction management in the construction industry is crucial [7]. Energy conservation and emissions reduction in the construction industry is of great significance. In this context, it is particularly important to analyze the current situation and influencing factors of regional construction carbon emission in China, which can provide a basis for the government to formulate relevant policies. Therefore, this problem has attracted extensive attention in the academic community [8].

With the rapid progress of urbanization in China and the rapid economic development of various regions [9], constrained by regional economic development, energy supply, economic, technological and other factors, the carbon emissions of the construction industry in each province are not independent. The continuous mutual flow and exchange of resources and information between regions inevitably leads to the existence of geospatial effects between neighboring regions, which strengthens the spillover effect of provincial construction carbon emissions and gradually strengthens the spatial correlation of construction carbon emissions in each region. In the deepening connection between regional trade and services [10] and the proposal of regional development strategies such as western development, the strategy of revitalizing northeast China and pioneer development in the east, carbon emissions show the characteristics of spatial heterogeneity, and there are obvious regional differences [11]. There is a complex and multi-threaded correlation between construction carbon emissions in various regions. A clear understanding of the spatial correlation characteristics and influencing factors of construction carbon emissions will help decision-making departments to take corresponding measures to implement construction carbon emission reduction management.

Research on carbon emission reduction management in the construction industry by scholars at home and abroad has mainly focused on the project and regional levels [12]. At the project level, the research on construction carbon emission reduction management mainly focuses on the measurement calculation method of construction carbon emissions and collaborative emission reduction management. Carbon emission measurement with the life cycle assessment (LCA) method and collaborative management of carbon emission reduction are carried out for different life cycle stages of buildings [13], building materials [14], building components and parts [15] and building structure forms [16], among other elements. The building design criteria should be promoted to provide reference for low-carbon building design and whole life cycle management. Thus, an optimized building design process that considers sustainability criteria in a holistic manner should be supported [17].

At the regional level, the research on construction carbon emissions has mainly focused on the calculation of construction carbon emissions and the study of influencing factors. Du et al. investigated the relationship between construction carbon emissions and economic growth with the Kuznets curve and elastic decoupling model based on the calculation of construction carbon emissions and found the main factors affecting its elastic decoupling [18]. Huo et al. established China's building energy consumption and emissions model, reasonably calculated the carbon emissions generated during building operation and put forward suggestions on energy conservation and low-carbon development in the construction industry [19]. Based on previous studies, Chen et al. studied the carbon footprint of the whole life cycle of buildings, analyzed the carbon emission sources of each stage and put forward corresponding carbon emission reduction strategies [20]. Zhang et al. proposed a Chinese building construction model based on process life cycle assessment to find out the main driving factors and influencing direction affecting building carbon emissions [21]. Tan et al. established the carbon emission prediction model system of China's construction industry and the model for assessing the potential of carbon emission reduction [22].

Due to local monetary development, electricity supply, economy, era and other elements, the carbon emissions of the construction industry in each province are different. The continuous flow and exchange of resources and information between regions will inevitably lead to geospatial effects between adjacent regions, strengthen the spillover effect

of provincial carbon emissions of the construction industry and gradually strengthen the spatial correlation of carbon emissions of the construction industry in each region [23]. The carbon emissions of the construction industry show significant spatial dependence [24]. At present, the research on the spatial correlation of construction carbon emissions is limited to "adjacent" or "similar" regions while ignoring the correlation between construction carbon emissions at the regional level. Social network analysis (SNA) is used to study the association between individuals in social networks. Because it takes into account the overall association of the network and the influence status of individuals in the network, it has been widely used in many disciplines [25]. Zhu et al. used the gravity model and social network analysis method to study the temporal and spatial variant characteristics of China's coal transportation network and discussed the status and role of each network node in the coal transportation network based on the coal output and input matrix data of China's provinces from 1990 to 2014 [26]. Yang et al. established China's industrial carbon emission efficiency network by the gravity model and social network analysis method to study the convergence and influencing factors of the regional ICEE of China's industrial carbon emissions [27]. In addition, the carbon emission intensity of the construction industry can more accurately reflect the development of the construction industry than the total carbon emissions. This is a deficiency in the research of carbon emissions in the construction industry in China [8]. Previous studies have focused on the management of construction carbon emission reduction at the project and regional levels. However, this does not consider the carbon emission intensity index, and it ignores the spatial characteristics of construction carbon emissions. Therefore, this research uses the gravity model and social network analysis method to calculate the influence degree of construction carbon emissions among provinces to construct the spatial network model for construction carbon emission intensity among the provinces in China, analyzing the influence statuses of provinces in the network and the correlation between provinces. The results provide a greatly significant basis for China to formulate and implement differentiated carbon emission reduction policies and provide an effective decision-making basis for building a more reasonable regional carbon emission reduction collaborative management mechanism to effectively achieve carbon emission reduction targets and achieve low-carbon sustainable development.

This paper contains five parts. A review of the state of research on the problem in question is provided in the introduction, followed by the analysis method's data sources in Section 2. The third part is an empirical analysis. Section 4 presents a discussion of the results. The research conclusions are given in Section 5, which is the final section of this article.

## 2. Materials and Methods

### 2.1. Carbon Emissions Estimation

At present, the main methods to calculate the carbon emissions of the construction industry include the emission coefficient method, input-output model, building life cycle management and measurement and material balance equation [28]. The Intergovernmental Panel on Climate Change (IPCC) carbon emission coefficient method is used to measure and calculate the carbon emissions of the construction industry in this investigation. The carbon emissions of the construction industry are divided into direct and indirect types. Direct carbon emissions mainly include the increase in carbon emissions caused by the consumption of fossil energy (e.g., raw coal, diesel, kerosene, briquette, gasoline, fuel oil, liquefied petroleum gas, lubricating oil and natural gas), electric power and thermal energy. Indirect carbon emissions are attributed to the consumption of building materials (e.g., cement, glass, steel, wood and aluminum) [29]. The carbon emissions of the construction industry can be calculated as follows:

$$CE_{i,j} = DCE_{i,j} + ICE_{i,j} \tag{1}$$

$$DCE_{i,j} = \left( \sum_{k=1}^{9} \left( FE_{i,j,k} \times fe_k \right) + TE_{i,j} \times te_j \right) \times \frac{12}{44} \times 10 - 6 + EE_{i,j} \times ee_i \tag{2}$$

$$ICE_{i,j} = \sum_{h=1}^{5} DE_{i,j,h} \times de_h{}^* (1 - a_h) \tag{3}$$

where, *CE*, *DCE* and *ICE* are the construction carbon emissions (tons), direct carbon emissions (tons) and indirect carbon emissions (tons), respectively, *FE*, *EE*, *TE* and *DE* are the energy consumption of fossils (tons), electric power (KW·h), heat (KJ) and building materials (kg), respectively, *fe*, *ee*, *te*, *de\** and ah denote the carbon emission factors of the fossil, electric power, heat, building, and recovery coefficients of the building materials, respectively, *i* is the region, *j* is the year, *k* is the type of fossil energy and *h* is the type of building materials, where 44/12 is the ratio of the molecular weights of $CO_2$ and C and $\alpha$ denotes the recovery coefficient of the building materials.

### 2.2. Social Network Analysis

The SNA method was developed on the basis of graph theory. It is a quantitative analysis method used to analyze "relational data" and to solve spatial network problems in economic systems, mainly including the overall network characteristics, individual network characteristics, structural hole analysis and spatial clustering analysis [30]. The spatial correlation network of the interprovincial carbon emission intensity is a collection to explore spatial carbon emission relationships [31]. Constructing a spatial association network is the foundation of social network analysis. Previous studies have concluded that the economic development level, population size and energy consumption are the main factors affecting carbon emissions [32]. Therefore, the gravity model was selected to determine the association relationship. The calculation formula of the modified gravity model is as follows:

$$y_{ij}^* = k_{ij} \cdot \frac{\sqrt[3]{P_i C_i G_i} \sqrt[3]{P_j C_j G_j}}{\left[ D_{ij} / \left( G_i - G_j \right) \right]^2}, \quad \text{where } k_{ij} = \frac{C_i}{C_i + C_j} \tag{4}$$

where $C_i$ and $C_j$ denote the construction carbon emission intensity of provinces *i* and *j* ($kg/10^4$ yuan), respectively, $G_i$ and $G_j$ denote the per capita gross domestic product (GDP) of provinces *i* and *j* in the current year, respectively, and the difference ($G_i - G_j$) is the variable of economic gap between the provinces *i* and *j*. $P_i$ and $G_i$ denote the proportion of the urban population and the GDP (CNY $10^4$) of province *i*, respectively, $P_j$ and $G_j$ denote the proportion of the urban population and the GDP (CNY $10^4$) of province *j*, respectively, $D_{ij}$ is defined as the geographic distance between provinces *i* and *j* and is expressed by the straight-line distance between the two provincial capitals in the research (km), $k_{ij}$ is the empirical carbon emission intensity constant and $y_{ij}^*$ indicates the degree to which the construction carbon emission intensity of province *i* is affected by province *j*. According to Equation (4), the gravitational matrix between different provinces was obtained, and each row of the gravitational matrix was averaged. Then, the row data in the gravitational matrix were compared with the average value, and if the row data were higher than the mean of each row in the gravitational matrix, they were marked as 1. This indicates that the province in which the row is located is associated with the carbon emission intensity in the province where the column is located. The line was directly linked to the column. Otherwise, it was recorded as 0, which indicates that there was no correlation. Therefore, the spatially correlated network matrix of construction industry's carbon emission intensity across provinces was established.

### 2.2.1. Indicators of the Overall Network

The network density, hierarchy, and efficiency were used to research the characteristics of the overall network of the construction carbon emission intensity. The overall network density indicates the closeness of the impact between provinces in the correlation network.

$$D = \frac{L}{N(N-1)} \qquad (5)$$

The network hierarchy is used to measure the asymmetric accessibility of the relationship between provinces in the associated network. A higher network hierarchy indicates that the hierarchical structure between provinces in the network is relatively unequal, and a few provinces are in a dominant position in the network, which should be balanced:

$$GH = 1 - \frac{R}{\max(R)} \qquad (6)$$

$$GE = 1 - \frac{M}{\max(M)} \qquad (7)$$

where $D$, $GH$ and $GE$ denote the network density, hierarchy and efficiency in the construction carbon emission intensity network, respectively, $N$ is the number of the points in the network, $L$ is the associations, $R$ and $\max(R)$ are the numbers of symmetrically researchable point pairs and the maximum possible numbers of symmetrically researchable point pairs, respectively, and $M$ and $\max(M)$ represent the numbers and maximum numbers of redundant lines, respectively.

### 2.2.2. Indicators of the Individual Network

The degree, closeness and betweenness are used to measure the structural characteristics of an individual network. The centrality of a point indicates the degree of association between the point and other points in the association network. The closeness of proximity to the center of a point indicates the degree to which the point is not affected by other points. Generally, the shortest distance between points is used for measurement. The higher the closeness of proximity to the center, the more correlation with other provinces the construction carbon emission of the point will have, and the more central in the network it will be. Betweenness measures the degree to which a point is in the middle of other points (i.e., the intermediary role of a point in the network). The degree, closeness and betweenness are as follows:

$$C_{RD} = \frac{n}{(N-1)} \qquad (8)$$

$$C_{APi} = \frac{n-1}{\sum\limits_{j=1}^{n} d_{ij}} \times 100 \qquad (9)$$

$$C_{ABi} = \frac{2\sum\limits_{j}^{n}\sum\limits_{k}^{n} b_{jk}(i)}{n^2 - 3n + 2}, \; j \neq k \neq i, \; \text{and} \, j < k, \, b_{jk}(i) = g_{jk}(i)/g_{jk} \qquad (10)$$

where $C_{RD}$, $C_{APi}$ and $C_{ABi}$ represent the degree, closeness and betweenness in the construction carbon emission intensity network, respectively, $n$ is the number of individuals directly associated with a point, $d_{ij}$ is the length of the shortcut from point $i$ to point $j$, $g_{jk}$ is the number of the shortcuts between provinces $i$ and provinces $k$ and $g_{ik}(i)$ is the number of shortcuts crossing province $i$, which are located in provinces $i$ and $k$.

In the social network, there will be no direct connection or intermittent relationship between some individuals. On the whole, it seems that there are holes in the network structure, as in structural holes. The effective size, efficiency, constraint and hierarchy are generally used to measure structural holes [33].

*2.3. Spatial Cluster Analysis*

The CONCOR model is the main method of spatial clustering analysis in social network analysis which can describe the characteristics of spatial clustering. The CONCOR model can analyze the role of each location (block) in a network. Through block model analysis, a new dimension of network characteristics can be revealed and described the internal structure state of a spatial correlation network [33]. Moreover, the number of blocks in the network, the provinces contained in each block and the relationship and connection mode between blocks can be analyzed. Block model analysis is a common analysis method in SNA, and the block model is mainly used to analyze the location of each node in the overall network, which is more intuitive for examining the development status and understanding the spatial correlation network and complex links between various blocks.

*2.4. Ordinary Least Squares Method*

The ordinary least squares method (OLS) is the most basic and most used method in econometrics which predicts the response variables through a series of prediction variables. The principle is very simple, which is to find some values that minimize the square sum of the difference between the actual value and the model valuation and take them as the parameter estimation value. The unknown data can be easily obtained by using the least squares method, and the sum of the squares of the errors between the obtained data and the actual data can be minimized [29]. The least squares method can be used for curve fitting, and some other optimization problems can also be expressed by the least squares method by minimizing the energy or maximizing the entropy. The method is as follows:

$$Y_t = \alpha + \beta X_t + \mu_t \; t = 1, 2, \ldots n \tag{11}$$

where $Y_t$ is the dependent variable, $X_t$ is the independent variable, $\alpha$ and $\beta$ are regression coefficients and $\mu_t$ is the random error term.

*2.5. Data Sources*

In this investigation, the samples were 30 provinces in China (excluding Tibet, Hong Kong, Macao and Taiwan) during 2006–2017. The geographical distance between provinces was represented by the spherical distance between provincial capitals, which was calculated by ArcGIS. The source consumption of the energy consumption and building material consumption of the construction industry were from the China Statistical Yearbook (2006–2017) and China Energy Statistical Yearbook (2006–2017), respectively. The GDP, population at the end of the year and per capita GDP of each province required in the gravity model were all from the China Statistical Yearbook (2006–2017). The coefficients of construction carbon emissions and recovery coefficients of construction carbon emissions were from previous studies [34]. According to Equations (1)–(3), the construction carbon emissions were calculated from 2006 to 2017, and the results are shown in Appendix A Table A1. Then, Equation (4) was used to measure the correlation between various provinces. The geographica l distance between provinces was represented by the spherical distance between provincial capitals, which was calculated by ArcGIS software. Finally, Ucinet software was used to establish the spatial correlation network of the construction industry.

**3. Results**

*3.1. Construction Carbon Emissions and Construction Carbon Emission Intensity*

From 2006 to 2017, the carbon emissions of China's construction industry increased from 74.3 million tons to 201.4 million tons, with an average annual growth rate of about 14%. The GDP of the construction industry increased from CNY 4150.8 billion to CNY 21,379.6 billion, with an average annual growth rate of about 34%. The provinces with large carbon emissions from China's construction industry are mostly in the southeast, extending from the eastern coastal areas to the west. The emissions of Zhejiang province

were more prominent, and the surrounding Fujian, Jiangsu and Hubei provinces also had high emissions, while the carbon emissions in the more remote areas in the north and south of China were mostly low.

In 2017, the carbon emissions of China's provincial construction industry showed a prominent trend in the southeast, while the surrounding areas were low, and the regional differences in the carbon emissions of the construction industry were significant (Figure 1). From 2006 to 2017, the carbon emission intensity of China's construction industry showed an inverted N-type variation trend. From 2006 to 2009, the carbon emission intensity of China's construction industry showed a downward trend as a whole. This is because in 2003, the State Council issued a document to establish China's real estate industry as the pillar industry of the national economy, and the output value of the construction industry increased rapidly. With the outbreak of the global economic crisis in 2008 [35], the development of the real estate industry was hit, the growth of the construction industry was slow, and the carbon emission intensity of the construction industry increased slightly. From 2010 to 2012, the carbon emission intensity of China's construction industry showed an obvious upward trend. During this period, China's real estate entered a relatively stable period of coordinated development. From 2013 to 2017, the carbon emission intensity of China's construction industry showed a downward trend. The tendency of carbon emissions exhibited differences, and the distributions of the regional economy and carbon emissions presented spatial heterogeneity [36].

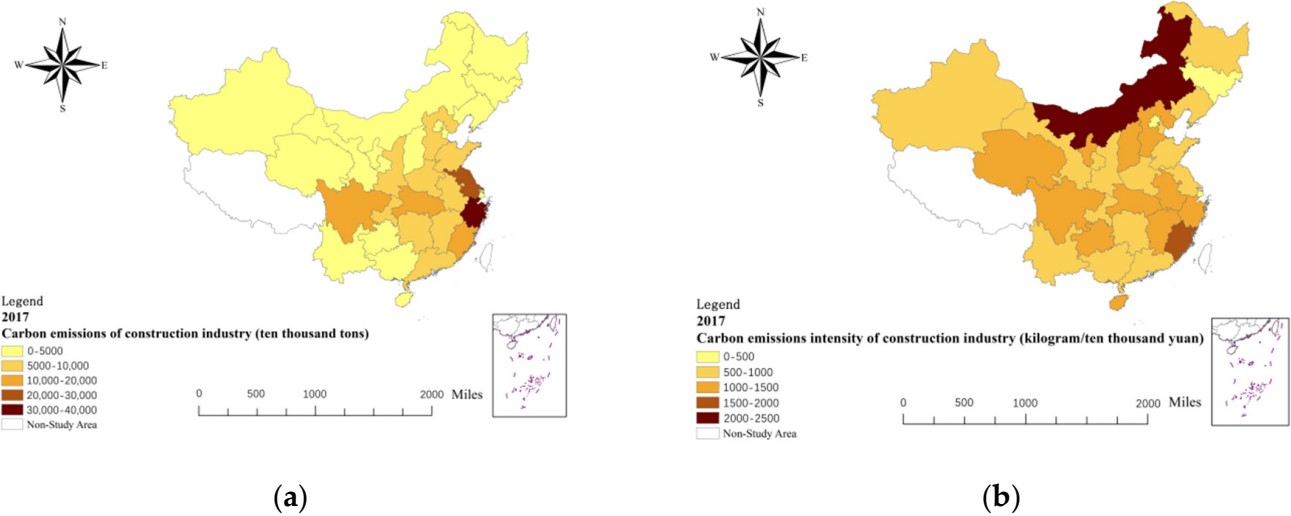

(**a**)                                                                                             (**b**)

**Figure 1.** Distribution map of provincial construction carbon emissions and emission intensity in 2017: (**a**) construction carbon emissions and (**b**) construction carbon emission intensity.

### 3.2. Spatial Network Analysis of Construction Carbon Emission Intensity

3.2.1. Overall Network Characteristics

According to the modified gravity model, a spatial correlation network of interprovincial construction carbon emission intensity was established. Figure 2 shows the typical spatial correlation network structure. The overall network characteristic index including the network density, network efficiency and network hierarchy were calculated according to Equations (5)–(7).

The overall network density of the carbon emission intensity of China's construction industry from 2006 to 2017 is shown in Figure 2. The maximum number of network correlations was 435. It can be seen that the number of correlation relations and the overall network density generally showed a gradual upward trend. With economic and construction development, the correlation of interprovincial carbon emissions tended to be close. The network density increased significantly from 2007 to 2011 and fluctuated downward in 2012. Since 2013, the network density has shown a slow downward trend. In

2012, due to the sudden increase in emissions in Jilin, Shandong and Hubei, they occupied a key position in the network and weakened the relationship between other provinces.

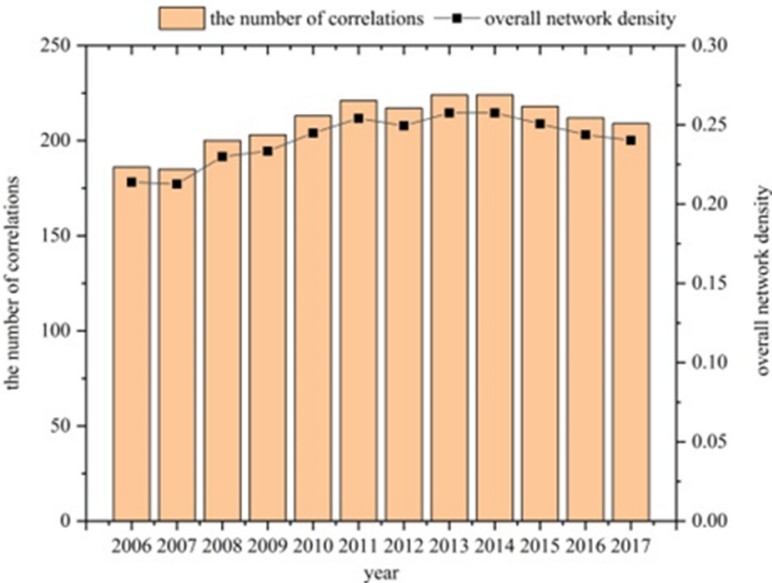

**Figure 2.** Overall network density and correlation of construction carbon intensity emission network.

After 2013, the construction carbon emissions of provinces and cities slowed down due to China's "Twelfth Five-Year Plan" [18], while the economic and productivity levels have still been improving steadily, narrowing the gap in construction carbon emission intensity between provinces and slightly reducing the degree of correlation impact of the provinces. Previous research has proposed that the level of economic activity is strongly correlated to carbon emissions [37].

From 2006 to 2017, the overall network level of the carbon emission intensity of China's construction industry showed a decreasing trend, as shown in Figure 3. With the development of China's overall production capacity, the strict hierarchical structure in the association network has been gradually broken. Some of the provinces that originally had a lower level of development made great economic progress, and the statuses of provinces and cities gradually became equal. The network hierarchy showed a significant decline in 2010, 2012 and 2016. The subprime mortgage crisis in 2008 had an impact on the original network statuses of provinces and weakened the dominant position of more developed provinces [35]. In addition, environmental protection policies in China have also been implemented, with the policies focused on limiting the construction carbon emissions of economically developed provinces and maintaining the overall fairness of the network. Since 2013, the network hierarchy of each province has gradually stabilized, with only a small fluctuation in 2016, making the status of each province in the network more equal. The reduction shows that the provinces that were originally in a dominant position began to have an influence, and the provinces with better economic development were easier for driving the common development of other provinces, which helped to slow down the overall construction carbon emissions.

3.2.2. Individual Network Characteristics

The degree, closeness and betweenness of the individual network characteristic indexes were calculated according to the Equations (8)–(10). The data of 2017 were calculated from the statistical yearbook data of the latest year, and the network density, network level and network efficiency of that year fluctuated compared with previous years. Therefore, this study selected the data from 2017 to study the individual network characteristics of carbon emission intensity of construction industry. The overall network showed an obvious "core-edge" state, and the provinces with more relationships in the middle were

Beijing, Tianjin, Shanghai, Jiangsu and Zhejiang, as shown in Figure 4. The economic development of these provinces is relatively rapid, and the level of production technology is very high. These provinces are closely related to the economic development of other provinces, and the carbon emission intensity of the construction industry is strongly related to other provinces [18]. The calculation of individual network indicators of carbon emission intensity of construction industry is shown in Table 1.

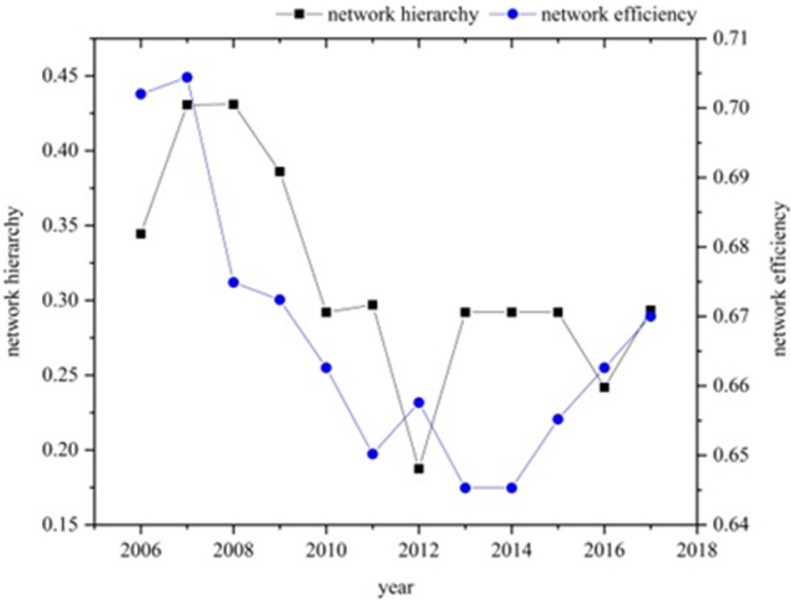

**Figure 3.** Network hierarchy and network efficiency of construction carbon emissions network.

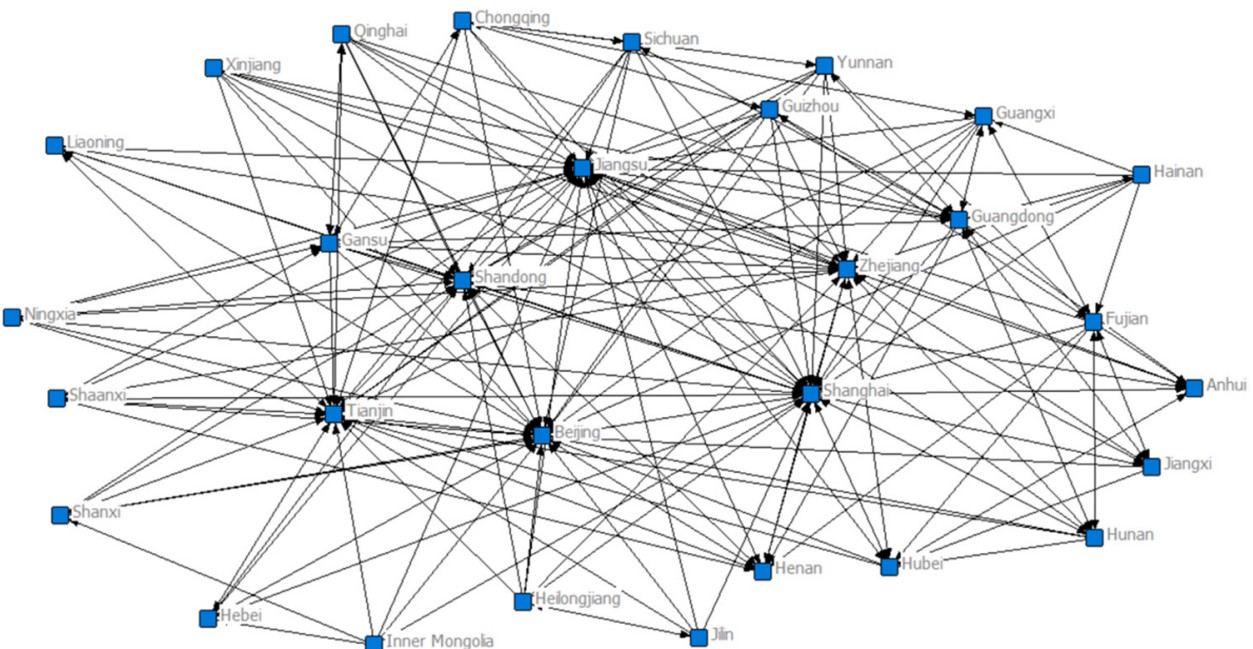

**Figure 4.** Spatial network of China's inter-provincial construction carbon emissions intensity in 2017.

**Table 1.** Analysis on the network centrality of inter-provincial construction carbon emissions intensity in 2017.

| Province | Degree Centrality | | | | Closeness Centrality | | Betweenness Centrality | |
|---|---|---|---|---|---|---|---|---|
| | Out-Degree | In-Degree | Centrality | No. | Centrality | No. | Centrality | No. |
| Beijing | 5 | 23 | 79.3 | 3 | 82.9 | 3 | 10.5 | 3 |
| Tianjin | 6 | 20 | 72.4 | 4 | 78.4 | 4 | 7.9 | 4 |
| Hebei | 4 | 5 | 20.7 | 26 | 55.8 | 26 | 0.1 | 24 |
| Shanxi | 5 | 4 | 20.7 | 26 | 55.8 | 26 | 0.1 | 24 |
| Inner Mongolia | 6 | 0 | 20.7 | 26 | 55.8 | 26 | 0.1 | 24 |
| Liaoning | 6 | 2 | 20.7 | 26 | 55.8 | 26 | 0.1 | 24 |
| Jilin | 5 | 1 | 17.2 | 30 | 54.7 | 30 | 0.0 | 30 |
| Heilongjiang | 7 | 1 | 24.1 | 19 | 56.9 | 19 | 0.2 | 18 |
| Shanghai | 9 | 25 | 93.1 | 1 | 93.5 | 1 | 14.6 | 1 |
| Jiangsu | 4 | 27 | 93.1 | 1 | 93.5 | 1 | 14.6 | 1 |
| Zhejiang | 7 | 18 | 65.5 | 5 | 74.4 | 5 | 5.6 | 5 |
| Anhui | 3 | 7 | 24.10 | 19 | 56.9 | 19 | 0.2 | 18 |
| Fujian | 8 | 6 | 37.9 | 8 | 61.7 | 8 | 1.2 | 8 |
| Jiangxi | 7 | 5 | 24.1 | 19 | 56.9 | 19 | 0.2 | 18 |
| Shandong | 8 | 15 | 58.6 | 6 | 70.7 | 6 | 3.6 | 6 |
| Henan | 6 | 9 | 31.0 | 10 | 59.2 | 10 | 0.5 | 12 |
| Hubei | 6 | 6 | 31.0 | 10 | 59.2 | 10 | 0.7 | 9 |
| Hainan | 8 | 4 | 27.6 | 16 | 58.0 | 16 | 0.3 | 16 |
| Guangdong | 11 | 10 | 48.3 | 7 | 65.9 | 7 | 2.5 | 7 |
| Guangxi | 7 | 4 | 31.0 | 10 | 59.2 | 10 | 0.6 | 11 |
| Hainan | 7 | 1 | 24.1 | 19 | 56.9 | 19 | 0.2 | 18 |
| Chongqing | 9 | 4 | 31.00 | 10 | 59.2 | 10 | 0.4 | 15 |
| Sichuan | 8 | 2 | 27.6 | 16 | 58.0 | 16 | 0.3 | 16 |
| Guizhou | 9 | 2 | 31.0 | 10 | 59.2 | 10 | 0.5 | 12 |
| Yunnan | 9 | 2 | 31.0 | 10 | 59.2 | 10 | 0.5 | 12 |
| Shaanxi | 7 | 1 | 24.1 | 19 | 56.9 | 19 | 0.1 | 24 |
| Gansu | 10 | 4 | 37.9 | 8 | 61.7 | 8 | 0.7 | 9 |
| Qinghai | 8 | 1 | 27.6 | 16 | 58.0 | 16 | 0.2 | 18 |
| Ningxia | 7 | 0 | 24.1 | 19 | 56.9 | 19 | 0.1 | 24 |
| Xinjiang | 7 | 0 | 24.1 | 19 | 56.9 | 19 | 0.2 | 18 |
| Mean value | 7.0 | 7.0 | 37.5 | - | 62.9 | - | 2.2 | - |

The average degree centrality of the network was 37.5 in the province spatial correlation network of China's construction carbon emission intensity. In Shanghai, Jiangsu, Beijing, Tianjin, Zhejiang, Shandong, Guangdong, Gansu and Fujian, the degree centrality of these provinces was higher than the national average, and there were many local correlation relationships in the spatial correlation network of construction carbon emissions. Among them, the degree centrality of Shanghai and Jiangsu were the highest, reaching 93.1. The degree centrality indicated that there was a close spatial correlation between the economic development of Shanghai and Jiangsu and other provinces [38]. There was a spillover effect of provincial construction carbon emissions. Most of these provinces are located in the Bohai Rim, Yangtze River Delta and Pearl River Delta, except for Gansu and Fujian. In terms of the in-degree results, the top five provinces were Jiangsu, Shanghai, Beijing, Tianjin and Zhejiang. These provinces are located in the eastern coastal area, with a developed economy and convenient transportation.

From the calculated out-degree results, the average out-degree value of all provinces in China was 7.0, and the gap between provinces was not obvious. Among them, the higher-ranking provinces were Guangdong, Gansu, Chongqing, Guizhou and Yunnan. These provinces were relatively affected by the construction carbon emissions of other provinces in the associated network. Most provinces are located in the middle and north of China, which is limited by logistics and hinders the development of the economy and productivity. Due to its superior geographical location and rapid economic development,

the eastern coastal area has a strong influence on the spatial correlation and spillover effect of building carbon emissions.

It can be seen from Table 1 that the mean value of the closeness centrality was 62.9. The closeness centrality of Shanghai, Jiangsu, Beijing, Tianjin, Zhejiang, Shandong and Guangdong was higher than the national average. These provinces are the centers of economy, commerce and culture in the county. They are in the central position in the spatial correlation structure network of construction carbon emission intensity and play the role of central actors in the country. Provinces with central leadership make use of their own advantages in resources, policies, technology and funds to send the correlation of the construction carbon emission intensity to the surrounding provinces, and this affects the economic and productivity development of other provinces. Provinces with closeness centrality values less than the average, such as Shanxi, Jilin and Inner Mongolia, play the role of marginal actors in the network, and their construction carbon emission intensity values will be affected by the surrounding provinces.

The province spatial correlation network of China's construction carbon emission intensity gradually presents a "center-edge" structure. According to the calculation, the average value of the betweenness centrality of the spatial correlation structure network was 2.2. The values of Shanghai, Jiangsu, Beijing, Tianjin, Zhejiang, Shandong and Guangdong exceeded the country's average. Shanghai and Jiangsu had the highest value of 14.6. As economically developed provinces, Shanghai and Jiangsu are at the core of the spatial correlation network of building carbon emissions and play the role of a "bridge". In 2017, the betweenness centrality of the spatial correlation network of the construction carbon emission intensity was 66.8, while the sum of the betweenness centrality of the top seven provinces accounted for 88.8% of the national total. These provinces strongly guide the spatial correlation of construction carbon emissions among other provinces in the spatial correlation network. The betweenness centrality of Inner Mongolia, Jilin, Shaanxi and Ningxia provinces was lower. The economic development speed of these provinces was slow, and it was not easy to have a strong transmission effect. It can be seen that the construction carbon emissions were restricted by the geographical location, population scale and economic development level of each province.

The structural hole of the provincial spatial correlation network of the carbon emission intensity of China's construction industry in 2017 is shown in Table 2. The larger the effective size of the individual in the network, the easier it is to have structural holes in the network, and the smaller the degree of repetition of the network [39]. The higher the efficiency of a specific province in the network, the greater the impact of the province on other provinces, and they can act more efficiently. The higher the level of carbon emission intensity of the construction industry, which indicates that the province is in the center of the network, it is more likely to become the leader and controller of the network. The provinces with an effective scale greater than the average value of 7.7 were Jiangsu, Shanghai, Beijing, Tianjin, Zhejiang, Guangdong, Shandong and Fujian. Most provinces have not yet formed a relatively perfect and effective relationship network, and few provinces have effective scale advantages in the associated network.

From the perspective of limitation, Jiangsu, Shanghai, Beijing, Tianjin, Shandong, Zhejiang, Guangdong, Gansu, Chongqing, Fujian, Guangxi, Guizhou, Yunnan, Henan, Sichuan and Hunan were below the average value of 0.4 among the 30 provinces. From the perspective of effective scale and limitation, the top seven provinces were Jiangsu, Shanghai, Beijing, Tianjin, Zhejiang, Guangdong and Shandong, which play an important role in the overall network and are in the structural hole of the network.

### 3.3. Spatial Cluster Analysis

The CONCOR method was used to analyze the spatial clustering characteristics of the network with the data of 2017, and the 30 provinces were divided into 4 blocks. In the province spatial correlation network structure of China's construction carbon emission intensity, there were 209 correlation relationships among the 4 blocks. There was an obvious

spatial correlation and spillover effect between the four blocks of the carbon emission intensity of the construction industry. In order to visually describe the relationship between blocks, the number of relationships within and between the blocks is shown in Figure 5.

**Table 2.** Structural hole index system of carbon emission intensity of construction industry in 30 provinces of China.

| Province | Effective Scale | Efficiency | Limit Degree | Grade Degree | Effective Scale Ranking | Limit Degree Ranking |
|---|---|---|---|---|---|---|
| Beijing | 19.8 | 0.9 | 0.2 | 0.3 | 3 | 3 |
| Tianjin | 17.7 | 0.8 | 0.2 | 0.3 | 4 | 4 |
| Hebei | 3.2 | 0.5 | 0.5 | 0.1 | 27 | 27 |
| Shanxi | 3.2 | 0.5 | 0.5 | 0.1 | 27 | 27 |
| Inner Mongolia | 3.2 | 0.5 | 0.6 | 0.0 | 27 | 29 |
| Liaoning | 3.9 | 0.7 | 0.5 | 0.1 | 24 | 26 |
| Jilin | 2.7 | 0.5 | 0.6 | 0.1 | 30 | 30 |
| Heilongjiang | 4.4 | 0.6 | 0.5 | 0.0 | 21 | 20 |
| Shanghai | 21.9 | 0.8 | 0.2 | 0.2 | 2 | 2 |
| Jiangsu | 22.2 | 0.8 | 0.2 | 0.2 | 1 | 1 |
| Zhejiang | 15.3 | 0.8 | 0.3 | 0.3 | 5 | 6 |
| Anhui | 3.7 | 0.5 | 0.5 | 0.1 | 26 | 25 |
| Fujian | 8.0 | 0.7 | 0.4 | 0.2 | 8 | 10 |
| Jiangxi | 4.7 | 0.7 | 0.4 | 0.1 | 19 | 18 |
| Shandong | 11.6 | 0.7 | 0.3 | 0.2 | 7 | 5 |
| Henan | 5.9 | 0.7 | 0.4 | 0.1 | 13 | 14 |
| Hubei | 5.0 | 0.6 | 0.4 | 0.1 | 16 | 17 |
| Hainan | 5.6 | 0.7 | 0.4 | 0.1 | 14 | 16 |
| Guangdong | 11.8 | 0.8 | 0.6 | 0.3 | 6 | 7 |
| Guangxi | 6.1 | 0.7 | 0.4 | 0.0 | 12 | 11 |
| Hainan | 4.3 | 0.6 | 0.5 | 0.1 | 22 | 22 |
| Chongqing | 5.0 | 0.6 | 0.3 | 0.0 | 16 | 9 |
| Sichuan | 5.6 | 0.7 | 0.4 | 0.0 | 15 | 15 |
| Guizhou | 6.5 | 0.7 | 0.4 | 0.1 | 10 | 12 |
| Yunnan | 6.5 | 0.7 | 0.4 | 0.1 | 10 | 12 |
| Shaanxi | 4.2 | 0.6 | 0.5 | 0.1 | 23 | 24 |
| Gansu | 7.1 | 0.6 | 0.3 | 0.0 | 9 | 8 |
| Qinghai | 4.7 | 0.6 | 0.4 | 0.1 | 18 | 19 |
| Ningxia | 3.7 | 0.5 | 0.5 | 0.0 | 25 | 23 |
| Xinjiang | 4.6 | 0.7 | 0.5 | 0.1 | 20 | 21 |

The internal relationship and spillover relationship of the four blocks were different. The internal connection of block 1 was the closest, and it accepted many spillover relationships from blocks 3 and 4, which illustrates that the construction carbon emission generated by its own development had a great impact on blocks 3 and 4, and there was also an impact relationship on the carbon emissions among the provinces within the block, with less impact on block 2. Most of the cities included in block 1 are located in the Yangtze River Delta, with strong geographical advantages, rapid economic development and large carbon emissions which are easy to spread to other provinces, especially the nearby areas with low development statuses. Block 2 contained few provinces and had no internal relationship, but there was a considerable number of sending and receiving relationships between the network and other plates, being in the position of a typical "broker". These provinces included in the plate are at the forefront of China's development status and play an "intermediary" role in the network. Connecting the provinces with a better development status with other provinces is conducive to balancing the status of each province in the network and reducing the gap in carbon emission intensity. The number of internal relations of block 3 was relatively large. Because the provinces included were not dominant in geographical location and low in economic level, the overall development speed was slow, and they were more affected by other blocks in the network, but they were easy to

relate internally. From the above analysis, it can be seen that the "broker" effect was significant in the spatial network of the construction carbon emission intensity, where blocks 2 and 4 played an intermediary role in the network. The construction carbon emissions of provinces with lower development levels would be significantly affected by developed areas. The "broker" plays a certain role, namely of a "bridge" in the network, and it can convey the impact relationship of construction carbon emissions and also alleviate the large grade gap between provinces. The "beneficiary" role of block 1 is also very important. It received a lot of spillover relations from other plates in the overall network. Therefore, it should maintain and make use of its advantage of a high economic development level to drive other provinces to jointly develop productivity and make the network structure more equal. From the above analysis, there are 3 type blocks in the network, and blocks 2 and 4 belong to the "broker" block.

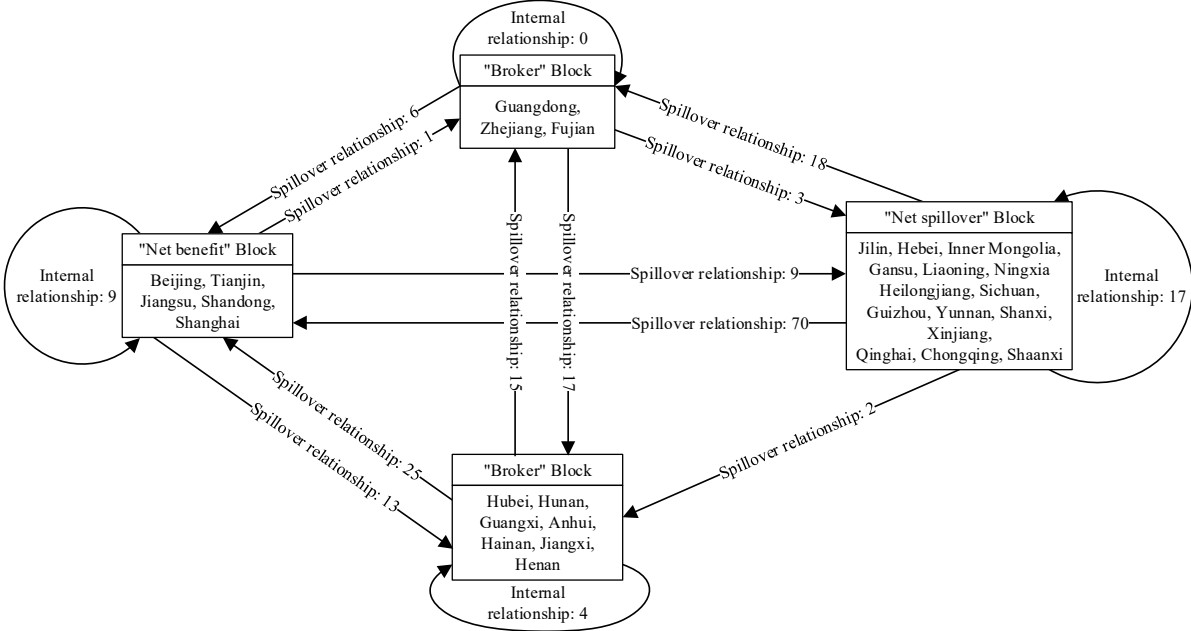

**Figure 5.** Correlation between the four parts of the associated network.

### 3.4. Carbon Emission Intensity Effect of the Construction Industry

The spatial correlation structure and characteristics of the construction carbon emission intensity are very important for analyzing the characteristics and transmission mechanism of construction carbon emissions. The driving effect of the construction carbon emission intensity was studied by econometric regression analysis. Several pre-analysis checks were performed on the panel data for analysis [40].

#### 3.4.1. Overall Spatial Network Structure Effect Analysis

The construction carbon emission intensity in each year and the standard deviation of the inter-province carbon emission intensity were selected as dependent variables, and the network density, network hierarchy and network efficiency of construction carbon emission intensity were used as dependent variables for regression analysis with the ordinary least squares (OLS) method. In order to eliminate the dimensional influence, all variables in the study used the natural logarithm, and the calculation results are shown in Table 3.

It can be seen from Table 3 that in Equations (1)–(3), the national construction carbon emission intensity was negatively correlated with the network density and network hierarchy and positively correlated with the network efficiency. The network density and network hierarchy were increased by 1%, and the construction carbon emission intensity would be reduced by 0.463% and 0.275%, respectively, and when the network efficiency was increased by 1%, the construction carbon emission intensity would be increased by 0.935%.

The reduction in the network efficiency of the construction carbon emission intensity means an increase in the number of effective connections in the network, which is conducive to reducing the difference in development level between provinces, making the network structure more stable and reducing the construction carbon emission intensity.

**Table 3.** Effect analysis results of the overall network structure.

| Dependent Variable | National Building Carbon Emission Intensity | | | Standard Deviation of Building Carbon Emission Intensity in a Province | | |
|---|---|---|---|---|---|---|
| **Model** | **(1)** | **(2)** | **(3)** | **(4)** | **(5)** | **(6)** |
| Constant | −9.476 *** | −9.140 *** | −8.436 *** | −1.955 | −12.742 *** | −14.235 *** |
| Network density | −0.463 | | | 5.346 | | |
| Network hierarchy | | −0.275 | | | −2.679 ** | |
| Network efficiency | | | 0.935 | | | −11.475 |
| R-squared | 0.009 | 0.040 | 0.007 | 0.145 | 0.463 | 0.129 |
| RESET test | 0.94 | 5.63 | 0.43 | 1.00 | 3.32 | 0.78 |
| | (0.4704) | (0.0279) | (0.7361) | (0.4486) | (0.0866) | (0.5433) |
| Breusch–Pagan test | 0.79 | 6.76 | 0.48 | 0.78 | 5.69 | 0.56 |
| | (0.3743) | (0.0093) | (0.4876) | (0.3776) | (0.0170) | (0.4560) |
| Breusch–Godfrey test | 2.774 | 4.21 | 2.809 | 0.791 | 3.065 | 0.813 |
| | (0.0958) | (0.0402) | (0.0937) | (0.3739) | (0.0800) | (0.3672) |

Note: *** and ** represent significance levels of 1% and 5%, respectively. Values in parentheses are the parameter P estimated value.

However, the regression coefficients of Equations (1)–(3) were not significant, the R-squared value was relatively small, and the regression results and corresponding analysis of Equation (2) were inconsistent with expectations, which was caused by the excessive growth of national carbon emissions in 2011 and 2012. The year 2011 was the first year of the 12th Five-Year Plan. In response to national policies, major construction enterprises have increased their efforts to develop real estate projects, resulting in the rapid development of the construction industry. In 2011, the GDP of construction enterprises exceeded CNY 1 billion for the first time, making the growth rate of national construction carbon emissions reach an extreme value and resulting in great fluctuations in the intensity of national construction carbon emissions over the years, and the fitting degree of the model was reduced. The model results can still reflect the impact direction of the overall network indicators for the national construction carbon emission intensity.

3.4.2. Individual Spatial Network Structure Effect Analysis

The construction carbon emission intensity of each province was selected as the dependent variable, and the degree centrality, closeness centrality, and betweenness centrality of the construction carbon emission intensity were used as dependent variables for regression analysis with the OLS method. In order to eliminate the influence of the dimension, all variables were calculated using the natural logarithm, and the results are shown in Table 4.

**Table 4.** Effect analysis results of individual network structure.

| Model | (7) | (8) | (9) |
|---|---|---|---|
| Constant | −9.355 *** | −9.619 *** | −9.109 *** |
| Degree centrality | −0.367 *** | | |
| Closeness centrality | | −1.401 *** | |
| Betweenness centrality | | | −0.030 |
| Fish test | | | 0.58 |
| Wald | 17.01 *** | 21.23 *** | |
| Hausman | 0.12 | 0.15 | 3.4 * |
| FE/RE | RE | RE | FE |

Note: *** and * represent the significance levels of 1% and 10%, respectively. FE = fixed effect, and RE = random effect.

Degree centrality and closeness centrality had a significant impact on the construction carbon emission intensity. The regression coefficient of Equation (9) was less significant due to the construction carbon emission intensity of Hebei, Jilin and Jiangsu provinces changing in an unstable way in 2011 and 2012. However, it still correctly indicated the impact direction of betweenness centrality on the construction carbon emission intensity. According to the regression coefficients of each model, when the centrality of the individual space network increased, it would reduce its own construction carbon emission intensity.

In order to study the impact of the structural hole characteristics of each province on its own construction carbon emission intensity, the carbon emission intensity was taken as the dependent variable, and the effective scale, efficiency, restriction degree and grade degree were taken as the independent variables for regression. All variables were standardized to eliminate the dimensional influence. According to the regression results in Table 5, the regression coefficients of the effective scale, efficiency, restriction degree and grade degree of the spatial correlation of the carbon emission intensity of the construction industry were −0.251, −0.218, 0.263 and −0.146, respectively, and this indicates that the network structure of the spatial correlation of the carbon emission intensity of the construction industry had a significant impact on the carbon emission level of the construction industry. After the efficiency was improved, some leading provinces had more influence in the network, retained "comparative advantages" in energy conservation and could drive themselves and surrounding provinces to jointly reduce the regional construction carbon emission intensity. At the same time, the central position of economically developed provinces in the network and the ability to control structural holes were strengthened, which is conducive to reducing carbon emissions from the construction industry.

**Table 5.** Effect analysis results of structural hole index.

| Explained Variable | Carbon Emission Intensity of Construction Industry | | | |
|---|---|---|---|---|
| Model | (10) | (11) | (12) | (13) |
| Constant | 0.346 *** | 0.370 *** | 0.148 * | 0.323 *** |
| Effective scale | −0.251 ** | | | |
| Efficiency | | −0.218 * | | |
| Restriction degree | | | 0.263* | |
| Grade degree | | | | −0.146 |
| R-squared | 0.136 | 0.120 | 0.124 | 0.034 |

Note: ***, ** and * represent the significance levels of 1%, 5% and 10%, respectively.

## 4. Discussion

As can be seen from Figure 1, the emission intensity of Inner Mongolia and Fujian Province was high in 2017. The emission intensity of most provinces was between 500 and 1500 kg/CNY 10,000. This indicates that China's overall emission intensity was in a relatively balanced state. The years 2011 and 2012 were the first two years of the 12th Five-Year Plan. China launched a series of policies to accelerate the transformation of the economic development mode, adjust the economic structure and promote steady and rapid economic development to improve the quality of China's overall economic development [18]. In addition, major construction enterprises also began to vigorously develop the real estate industry, thus accelerating the development speed of the construction industry and increasing the carbon emissions of the construction industry to a great extent [41]. In 2011, the consumption of cement materials in Hebei and Jiangsu increased suddenly, which greatly increased the total emissions in 2011, while the consumption of cement in Jilin increased from 101,950 million tons in 2011 to 114,964 million tons in 2012, resulting in a sharp increase in the total emissions in 2012. After 2013, the Chinese government strongly advocated for the development of resource-saving cities guided by energy conservation and emission reductions [35].

With the increase in network density, the relationship between the provincial construction carbon emissions was closer, and it was easier for economically developed provinces to drive other provinces to develop together, to improve the overall level of economic

development and to reduce the intensity of national construction carbon emissions. The agglomeration of construction industries and the cross-regional flow of energy and talent will inevitably promote network density in the spatial correlation of carbon emissions in different provinces. Thus, China still has a big gap between the largest possible network association numbers. From 2006 to 2017, the overall network efficiency of the carbon emission intensity of China's construction industry showed a downward trend, but there was a certain fluctuation in the downward process. It can be seen that the trend of change in the network efficiency corresponded to the network density, and the overall network tended to be stable. The network efficiency fluctuated upward in 2012. Due to the inadequate carbon emission control, the total emission in that year was much higher than that in other years, even reaching the peak during the development of the construction industry. However, the trend of change in carbon emissions was inconsistent with its economic development level, resulting in greater emission correlation to the surrounding provinces. The tendency reduced the provincial correlation degree of the entire network and improved the network efficiency. The annual carbon emission intensity of each province has decreased steadily since 2014, which has been effectively controlled, and the network efficiency has also shown a slight upward trend. The reduction in the network efficiency of the construction carbon emission intensity means an increase in the number of effective connections in the network, which is conducive to reducing the difference of development level between provinces, making the network structure more stable and reducing the construction carbon emission intensity. In the spatial network of the carbon emission intensity of the construction industry, the provinces with lower restrictions are in an open position in the network, having a strong ability to control other provinces, and there will be more structural holes. The entire network has a significant spatial association effect, and it has no isolated individuals [42].

The spatial fairness of the carbon emission intensity of the construction industry is an important factor affecting construction carbon emission reduction and regional differences in policies of construction carbon emission reduction [18]. The network hierarchy and network efficiency were increased by 1%, and the standard deviation of the construction carbon emission intensity would be reduced by 2.7% and 11.5%, respectively, while the network density increased by 1%, and the standard deviation of the construction carbon emission intensity would be increased by 5.3%. Due to the sharp increase in building emissions in Hebei, Jiangsu, Jilin and other provinces in 2011 and 2012, the standard deviation of the construction carbon emission intensity fluctuated greatly, resulting in low significance for the model. Therefore, reasonable allocation and adjustment of China's province construction carbon emission spatial network structure will help to reduce China's construction carbon emission intensity and narrow the difference for the province construction carbon emission intensity [34].

The top seven provinces in degree centrality were Shanghai, Jiangsu, Beijing, Tianjin, Zhejiang, Shandong and Guangdong, which were several provinces with more local construction carbon emission spatial relationships in the spatial network. These provinces are located on the eastern coast and cover economically developed areas, such as "Beijing Tianjin Hebei", the "Yangtze River Delta" and the "Guangdong Hong Kong Macao Bay Area". These provinces have more exchanges and flows in economy, talents and technology between various regions. In addition, talent flow, technology exchange, and the rapid development of economic integration have reduced the spatial difference of construction carbon emissions, and these led to the reduction in the construction carbon emission intensity [43]. The provinces with higher intermediary centrality play a stronger "intermediary" role in the spatial network structure of construction carbon emissions, and they help to adjust the direction and degree of the carbon emissions impact among provinces, strengthen the leading role of economically developed provinces in other provinces and finally reduce the overall construction carbon emission intensity. In order to reduce the intensity of construction carbon emissions, this should be managed as follows:

1. Establish a unified and standardized carbon emission statistical accounting system to improve building energy efficiency. Actively participate in the research on in-

ternational carbon emission accounting methods, enhance the capacity building of carbon emission statistical accounting, deepen the research on accounting methods and promote the establishment of a more fair and reasonable carbon emission accounting system [44]. Promote green low-carbon building materials and green construction methods, and promote the recycling of building materials, strengthen green design and green construction management, advocate for the concept of green low-carbon planning and design and enhance the climate resilience of urban and rural areas [45]. Consider the use of renewable energies and clean energies and decrease the non-renewable energies in the construction industry. Accelerate the update of standards for building energy conservation and municipal infrastructure and improve the requirements for energy saving and carbon reduction. Strengthen the R&D and promotion of energy-saving and low-carbon technologies applicable for different climate zones and different building types and promote the large-scale development of ultra-low energy consumption buildings and low-carbon buildings. Accelerate the energy-saving transformation of residential and public buildings and continue to promote the energy-saving and carbon reduction transformation of municipal infrastructure, such as old heating pipe networks. Accelerate the optimization of the building energy consumption structure, promote energy-saving technology and equipment and carry out the construction of an energy management system to achieve energy saving and efficiency.

2. Pay attention to the spatial correlation of provincial construction carbon emissions, optimize and adjust the spatial network structure of interprovincial construction carbon emissions, improve the spatial allocation efficiency of construction carbon emissions and realize spatial collaborative carbon emission reductions to realize new regionalism development [46]. The enhancement of the correlation with China's construction carbon emission intensity enables the country to coordinate the actions of all parties under the overall ideas of a national chessboard, forming a 1 + 1 > 2 emission reduction effect. The strengthening of the correlation with the construction carbon emission intensity requires further strengthening of regional spatial, economic and technological relationships.

3. Implement construction carbon emission reductions according to local conditions and strengthen the overall carbon emission reductions. Different levels of economic development lead to great differences in construction carbon emission intensity and uneven spatial distribution. Location factors should be taken into account when formulating collaborative emission reduction regulations [47]. In line with the distribution traits of construction carbon emissions and the situation of the province, the administration should construct an advanced demonstration area focusing on the Yangtze River Delta, Pearl River Delta and Beijing-Tianjin-Hebei Urban Agglomeration, drive the surrounding provinces through radiation and interaction among cities and link the carbon emission reduction management of adjacent regions. Fully consider the block structure characteristics of the spatial correlation network of construction carbon emissions, formulate regional differentiated construction carbon emission reduction policies and implement the classified management of construction carbon space. Due to the different statuses and roles of China's regions in the construction carbon emission intensity correlation network, the dependence of regional carbon emissions should be fully considered when formulating and allocating emission reduction tasks. Make use of the block's own technical advantages and management experience, adjust the ideas of the block's economic and social development, promote the transformation of the economic development mode and implement the management of construction carbon emission reductions according to local conditions.

4. Give full play to the positive role of the overall network structure and individual network structure of the spatial correlation of construction carbon emissions in reducing the intensity of construction carbon emissions and enhancing the spatial equity of construction carbon emissions. Accelerate the establishment of the construction car-

bon emission reduction market system, further play the role of the market in resource allocation and continuously narrow the gap in the economy, resources and technology among provinces in the spatial correlation network of construction carbon emissions to promote the reduction of the overall construction carbon emission intensity. Improve the stability of the construction carbon emission spatial network structure, give play to the spillover effect of construction carbon emissions and continuously improve the equity of the construction carbon emission intensity and spatial emissions under the condition of generally reducing the emission intensity of each province in the construction carbon emission spatial correlation structure network [48]. Then, while maintaining the current emission reduction level, promote the development of the construction industry and effectively reduce construction carbon emissions by increasing the industry output value and developing green building technology [49].

Of course, there are cetrtain limitations in this paper. We only considered the economic, population and carbon emission variables in a modified gravity model and neglected the influences of other factors on the spatial correlation network of the construction carbon emission intensity. In addition, the distance between various provinces was only considered as the geographical distance between various provinces.

## 5. Conclusions

Based on the measurement of China's construction carbon emissions from 2006 to 2017, this paper constructed the spatial correlation network of the construction carbon emission intensity by using the modified gravity model. This paper analyzed the structural characteristics of the spatial correlation network of the construction carbon emission intensity at the provincial level and then analyzed its driving effect. The main conclusions are as follows:

1.  China's construction carbon emissions increased from 74.3 million tons in 2006 to 201.4 million tons in 2017, with an average annual growth rate of about 14%. In the process of development, China's construction carbon emissions were mainly concentrated in Zhejiang, Jiangsu, Hebei and other provinces, accounting for nearly half of the country's total emissions. However, Ningxia, Qinghai, Hainan and other provinces ranked last in the construction carbon emissions, and the total emissions of the last 10 provinces were less than 10% of the whole country, showing significant regional differences. The carbon emission intensity of the construction industry showed a fluctuating trend.

2.  The network density of the interprovincial spatial correlation of China's construction carbon emission intensity showed an increasing trend annually. The network hierarchy of the spatial correlation of the construction carbon emission intensity in China showed a downward trend, and the hierarchical network structure was gradually broken. The network efficiency generally showed a downward trend. There were small fluctuations up and down, and the spatial correlation network was more complex and stable. Shanghai, Jiangsu, Beijing, Tianjin, Zhejiang, Shandong and Guangdong were at the forefront of the standardized degree centrality, closeness centrality and betweenness centrality, with values that were higher than the national average. They play a core role in the spatial association network of construction carbon emissions and can also act as a "bridge".

3.  The results of the space block model analysis showed that the first block was composed of Beijing, Tianjin, Jiangsu, Shandong and Shanghai. The first block was in the "net benefit" in the spatial correlation structure model of construction carbon emissions, while Guangdong, Zhejiang and Fujian belonged to the second sector and were in the "broker" position in the spatial correlation structure model of construction carbon emissions. Jilin, Hebei, Inner Mongolia, Gansu, Liaoning, Ningxia, Heilongjiang, Sichuan, Guizhou, Yunnan, Shanxi, Xinjiang, Qinghai, Chongqing and Shaanxi beloned to the third block, and this block was in the "net overflow" in the spatial correlation structure model of construction carbon emissions, while Hubei,

Hunan, Guangxi, Anhui, Hainan, Jiangxi and Henan belonged to the fourth block and were in the "broker" position in the spatial correlation structure model of construction carbon emissions.

4. The indicators such as the network density, network hierarchy and network efficiency had an impact on the national construction carbon emission intensity, and the indicators such as the point degree and centrality would also significantly affect the construction carbon emission intensity of each province. The spatial correlation network structure of the construction carbon emission intensity would also affect the difference in the construction carbon emission intensity. The individual spatial network structure had a significant impact on the construction carbon emission intensity. The province correlation in the network would be closer with the degree centrality, closeness centrality and betweenness centrality increasing, which resulted in the reduction in the construction carbon emission intensity. The spatial correlation network of construction carbon emissions gradually developed a "center-edge" structure in the process of development.

In the future, it is hoped that a reasonable method to construct the spatial correlation network of the construction carbon emission intensity can be constructed.

**Author Contributions:** Methodology, Z.W. and T.W.; software, N.Z.; validation, Z.W., N.Z. and T.W.; investigation, Y.Z. and Z.Z.; data curation, Z.Z.; writing—original draft preparation, Z.W.; writing—review and editing, T.W.; funding acquisition, T.W. and N.Z. All authors have read and agreed to the published version of the manuscript.

**Funding:** This research was funded by the National Natural Science Foundation of China, grant numbers 72074034, 72001035 and 71871235, and the Fundamental Research Funds for the Central Universities, grant numbers 2020CDJSK01PY16 and 2021CDSKXYGG013.

**Institutional Review Board Statement:** Not applicable.

**Informed Consent Statement:** Informed consent was obtained from all subjects involved in the study.

**Data Availability Statement:** The data presented in this study are available on request from the corresponding author.

**Conflicts of Interest:** The authors declare no conflict of interest.

## Appendix A

**Table A1.** Total carbon emissions ($10^4$ tons) from the construction industry (2006–2017) for 30 provinces in China.

| | 2006 | 2007 | 2008 | 2009 | 2010 | 2011 | 2012 | 2013 | 2014 | 2015 | 2016 | 2017 |
|---|---|---|---|---|---|---|---|---|---|---|---|---|
| Beijing | 1770 | 2110 | 2416 | 3434 | 4033 | 4137 | 3379 | 3751 | 3866 | 3712 | 3576.6 | 3760.8 |
| Tianjin | 980 | 1398 | 1545 | 2079 | 2131 | 2858 | 2810 | 3699 | 5013 | 3272 | 2937.2 | 2569.6 |
| Hebei | 5389 | 2998 | 4256 | 4450 | 11,422 | 50,016 | 48,166 | 17,938 | 7829 | 8166 | 5961.4 | 6207.0 |
| Shanxi | 1579 | 1920 | 2940 | 2990 | 4517 | 3203 | 3262 | 3359 | 3899 | 3388 | 3417.1 | 3609.4 |
| Inner Mongolia | 881 | 1203 | 2581 | 1792 | 2165 | 2086 | 1923 | 1768 | 1745 | 1807 | 2073.62 | 2725.4 |
| Liaoning | 2327 | 2650 | 3553 | 4856 | 6181 | 10,019 | 8163 | 14,803 | 14,600 | 4960 | 4634.72 | 2721.0 |
| Jilin | 889 | 901 | 1183 | 1537 | 1576 | 1758 | 94,576 | 6687 | 7109 | 2127 | 1176.0 | 1086.3 |
| Heilongjiang | 738 | 853 | 1028 | 1127 | 1352 | 1737 | 1545 | 1605 | 1673 | 1275 | 1240.3 | 1158.5 |
| Shanghai | 2119 | 2132 | 2246 | 2409 | 2516 | 2659 | 2456 | 2538 | 2631 | 2270 | 2373.0 | 2640.2 |
| Jiangsu | 8350 | 10,597 | 13,503 | 13,897 | 15,819 | 64,244 | 28,216 | 21,570 | 23,470 | 21,322 | 20,803.5 | 20,985.1 |
| Zhejiang | 13,507 | 14,033 | 17,111 | 18,118 | 20,850 | 25,529 | 27,275 | 30,853 | 31,709 | 31,148 | 31,632.6 | 33,774.3 |
| Anhui | 1857 | 2381 | 2648 | 3073 | 4194 | 4535 | 4524 | 5564 | 6091 | 5169 | 6490.9 | 8293.7 |
| Fujian | 2557 | 2129 | 3849 | 4761 | 6008 | 5977 | 7442 | 9910 | 12,981 | 13,110 | 15,165 | 19,159.0 |
| Jiangxi | 1357 | 1271 | 1469 | 1734 | 1946 | 3080 | 3129 | 4071 | 2057 | 5301 | 5272.9 | 6768.6 |
| Shandong | 5149 | 4302 | 6481 | 7840 | 7907 | 7858 | 18,160 | 9577 | 9938 | 8919 | 8064.1 | 8078.9 |
| Henan | 2692 | 3904 | 4424 | 5443 | 6774 | 6909 | 8474 | 8066 | 22,227 | 6450 | 8439.4 | 8918.2 |
| Hubei | 3216 | 3689 | 3251 | 3907 | 4083 | 8313 | 15,027 | 13,767 | 16,232 | 13,431 | 16,902.4 | 14,107.0 |

**Table A1.** *Cont.*

| | 2006 | 2007 | 2008 | 2009 | 2010 | 2011 | 2012 | 2013 | 2014 | 2015 | 2016 | 2017 |
|---|---|---|---|---|---|---|---|---|---|---|---|---|
| Hunan | 3603 | 3865 | 4252 | 5132 | 5601 | 5296 | 6137 | 6689 | 7104 | 7340 | 7156.0 | 7686.0 |
| Guangdong | 3794 | 3886 | 3706 | 4166 | 5296 | 8236 | 3243 | 6935 | 7649 | 6766 | 5573.5 | 8326.7 |
| Guangxi | 984 | 997 | 1093 | 1417 | 1693 | 1927 | 2602 | 2111 | 2343 | 1697 | 2669.9 | 3159.3 |
| Hainan | 107 | 121 | 183 | 236 | 244 | 395 | 399 | 502 | 366 | 274 | 311.1 | 351.3 |
| Chongqing | 1646 | 2013 | 2683 | 2707 | 4597 | 4749 | 4353 | 5220 | 5426 | 5304 | 5583.9 | 5413.9 |
| Sichuan | 2962 | 3374 | 3869 | 4874 | 10,330 | 12,583 | 18,553 | 19,713 | 21,843 | 9089 | 9260.0 | 13,181.4 |
| Guizhou | 626 | 717 | 801 | 1166 | 865 | 1256 | 1491 | 2625 | 3205 | 4066 | 7381.6 | 3391.7 |
| Yunnan | 1302 | 1217 | 1430 | 1674 | 2275 | 2009 | 2467 | 5516 | 6118 | 2951 | 3091.2 | 3356.7 |
| Shaanxi | 1426 | 1888 | 3131 | 3273 | 3691 | 6099 | 4126 | 4449 | 5079 | 4940 | 5349.8 | 5512.8 |
| Gansu | 844 | 638 | 1316 | 942 | 957 | 2027 | 1459 | 2016 | 2194 | 1640 | 1992.9 | 1794.8 |
| Qinghai | 129 | 203 | 303 | 416 | 586 | 349 | 374 | 398 | 426 | 406 | 430.6 | 499.2 |
| Ningxia | 249 | 275 | 364 | 420 | 500 | 619 | 587 | 783 | 859 | 582 | 557.6 | 572.6 |
| Xinjiang | 1002 | 718 | 818 | 844 | 1092 | 3373 | 1821 | 1755 | 2330 | 1760 | 1588.4 | 1595.1 |

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
