# Peer review of "Spatial Correlation Network and Driving Effect of Carbon Emission Intensity in China’s Construction Industry"

_buildings, doi:10.3390/buildings12020201_

Round 1
Reviewer 1 Report
See the attached document.

Author Response
Thank you very much for your advice on our manuscaipt. We revised the manuscript accroding to your kind advice and resubmitted new version, and colored in the red.

Reviewer 2 Report
Dear Authors,
the content of the research addressed is highly relevant. In particular, research focused on emission reduction in the top emitting countries worldwide needs to be supported and advanced.
To improve the article, I recommend expanding the "Materials and Methods" section, as some of the methods mentioned are not described. Transparent description of all methods and used data will ensure reproducibility of results and furthermore understanding of the article.
Please see the attached word file for my detailed comments on the manuscript.
All the best for your publication.

Author Response
Thank you very much for your advice on our manusccript. we have revised the manuscript by your advice, and colored in the red.

Reviewer 3 Report
This research uses the gravity model, being focused on the carbon emission intensity index and the spatial characteristics of construction carbon emission. The study is interesting in the field of decarbonization of the construction sector. Suggestions are made below:
- It is suggested to shorten sentences. For example the sentence “To explore the spatial network structure characteristics and driving effects of carbon emission intensity in China's construction industry, the investigation combined the modified gravity model and social network analysis method to deeply analyze the spatially associated network structure characteristics and driving effects of carbon emission intensity in China's construction industry, based on the measurement of carbon emission data of China's construction industry from 2006 to 2017” is very long.
- It is suggested to cite the official source. For example, the statement “To address global climate change issue, the Chinese government has proposed to achieve peak CO2 emissions around 2030 and carbon neutralization by 2060 [10]” is supported by an article, rather than by the official Chinese government document where the decarbonization targets are intended.
- Check quote 27, the text refers to the assessment of carbon emissions in buildings, when it refers to a study of CO2 emissions at provincial scale.
- Check quote 28, the text refers to the assessment of carbon emissions in materials, when it refers to a study of CO2 emissions in building components (slabs).
- Check quotes 29, 30
- Citation 31 does not refer to a structural system.
- Lines 86,87, avoid repetition “..studied the carbon footprint of the whole life cycle of buildings which divided the whole life cycle of buildings into mining and production stages”
- Lines 97 avoid informal language “…in each province isn’t always…”
- Check quote 38
- Lines 108-112, It cites a study by Zhu et al., when citation [46] refers to a study by Daniyar et. al.
- A priori, it is strange that Carbon emission estimation methods have been supported by a study on prefabricated systems [48]
- p4, Line 144, Correct variable FE, called EF in the equation (2).
- Equation 4, the variables Pj and Gj are still to be defined.
- Equation 4, the units of measurement are missing for those variables that have them (e.g. geographic distance).
- Reference Figure 1 in the text before presenting it.
- Line 242, check “began to have influence” is repeated.
- p10, Line 326, Present Table 2 after mentioning it in the text.
- p14, `Lines 500-506, sentence long, check “Lead” after coma
- p14, Lines 520, consider “use of benchmarks”. Consider that rather than recycling, it is more effective in reducing CO2 emissions, reducing the use of natural resources and preventing construction waste. In addition, consider including the use of renewable energy vs versus non-renewable energies as another recommendation in the discussion section.
- In the discussion section, indicate limitations of the study, e.g., variables that have not been taken into account, or the validity of the results in terms of the hypotheses assumed.
- In conclusions, add future developments or lines to be carried out after this study to continue advancing knowledge.
Author Response
Thank you very much for your advice on our manuscript. we have revised the manuscript according to your kind advice and resubmitted new version.

Round 2
Reviewer 2 Report
Dear Authors,
thank you for your extensive revisions and replies to my comments.
Pls. correct the typo in source [31].
Attached detailed information for source [33]: https://www.scopus.com/record/display.uri?eid=2-s2.0-85063938864&origin=inward&txGid=8f8a2e082763fc7e17747a00b1a9aece
Good luck for publication!
Author Response
Thank you very much for your advice on the manuscript. We have revised it and resubmitted new version.
